# Identification of Redeye, a new sleep-regulating protein whose expression is modulated by sleep amount

Mi Shi[1], Zhifeng Yue[1], Alexandre Kuryatov[2], Jon M Lindstrom[2], Amita Sehgal[1,2]*

[1]Howard Hughes Medical Institute, University of Pennsylvania, Philadelphia, United States; [2]Department of Neuroscience, Perelman School of Medicine, University of Pennsylvania, Philadelphia, United States

**Abstract** In this study, we report a new protein involved in the homeostatic regulation of sleep in *Drosophila*. We conducted a forward genetic screen of chemically mutagenized flies to identify short-sleeping mutants and found one, *redeye* (*rye*) that shows a severe reduction of sleep length. Cloning of *rye* reveals that it encodes a nicotinic acetylcholine receptor α subunit required for *Drosophila* sleep. Levels of RYE oscillate in light–dark cycles and peak at times of daily sleep. Cycling of RYE is independent of a functional circadian clock, but rather depends upon the sleep homeostat, as protein levels are up-regulated in short-sleeping mutants and also in wild type animals following sleep deprivation. We propose that the homeostatic drive to sleep increases levels of RYE, which responds to this drive by promoting sleep.

## Introduction

Sleep is a common and prominent behavior in almost all vertebrate animals and also in most invertebrates (*Cirelli, 2009*; *Sehgal and Mignot, 2011*). The function of sleep is a mystery, but it is surely of great importance to animals, as prolonged sleep deprivation can lead to death. Anatomical studies in mammals and birds have revealed brain structures and neurotransmitters that regulate sleep and wakefulness (*Saper et al., 2010*). However, our understanding of the molecular mechanisms that drive the need to sleep is still in its infancy, partially due to the challenge of performing genetic experiments with mammalian models. In the past decade, several premier genetic organisms have been introduced into the sleep field, including fruit flies, worms and zebra fish. Mammalian counterparts of some sleep components identified in these model animals also regulate sleep (*Joho et al., 2006*), which argues that (i) behavioral genetics in lower organisms provides an efficient tool to identify sleep components, (ii) at least some of the mechanisms underlying sleep are conserved through evolution.

A two-process model for sleep regulation has been widely accepted by the sleep field. Process C (the circadian clock) controls the timing, in other words the onset and offset of sleep, whereas process S (the sleep homeostat) regulates sleep duration based on the sleep pressure built up during prior wakefulness (*Borbely, 1982*). This simple model explains sleep related phenomena, including sleep rebound after sleep deprivation. Molecular mechanisms of circadian control have been well characterized (*Zheng and Sehgal, 2012*), but, as noted above, relatively little is known about process S. Forward genetic screens in *Drosophila* have led to the identification of several mutants with altered sleep length, but while the genes implicated by these mutants are required for implementation of sleep drive, they have not yet been directly linked to this drive (*Sehgal and Mignot, 2011*).

Using a forward genetic screen, we identified a new sleep mutant we termed *redeye* (*rye*), which is directly controlled by the homeostatic drive to sleep. The *rye* mutation maps to a nicotinic acetylcholine receptor (nAChR), which interacts with a previously identified sleep-regulating protein, SLEEPLESS (SSS). Levels of RYE are expressed cyclically, in conjunction with the sleep state, and reflect sleep need

*For correspondence: amita@
mail.med.upenn.edu

**Competing interests:** The authors declare that no competing interests exist.

**Reviewing editor**: Leslie C Griffith, Brandeis University, United States

**eLife digest** Almost all animals need to sleep, including most insects. In experiments in the 1980s, a group of rats that were completely deprived of sleep died within only a few weeks. Sleep has been implicated in processes including tissue repair, memory consolidation and, more recently, the removal of waste materials from the brain. However, a full understanding of why we sleep is still lacking.

As anyone who has experienced jetlag can testify, the timing of the sleep/wake cycle is governed by the circadian clock, which leads us to feel sleepy at certain points of the day–night cycle and alert at others. The duration of sleep is regulated by a second process called sleep/wake homeostasis. The longer we remain awake, the more the body's need for sleep—or 'sleep drive'—increases, until it becomes almost impossible to stay awake any longer. Whereas many components of the circadian clock have been identified, relatively little is known about the molecular basis of this second process.

Now, Shi et al. have identified a key component of the sleep/wake homeostatic system using the fruit fly and genetic model organism, *Drosophila*. Flies with a mutation in one particular gene, subsequently named *redeye*, were found to sleep only half as long as normal flies. While the insects were able to fall asleep, they would wake again only a few minutes later.

*Redeye* encodes a subunit of a receptor that has previously been implicated in the control of wakefulness, known as the nicotinic acetylcholine receptor. Mutant flies had normal circadian rhythms, suggesting that their sleep problems were the result of disrupted sleep/wake homeostasis. Consistent with this, levels of redeye showed two daily peaks, one corresponding to night-time sleep and the second to the time at which flies would normally take an afternoon siesta. This suggests that redeye signals an acute need for sleep, and then helps to maintain sleep once it is underway.

While redeye is not thought to be the factor that triggers sleep per se, it is directly under control of the sleep homeostat, and may be a useful biomarker for sleep deprivation. The fact that *redeye* was identified in fruit flies, a species whose genome has been fully sequenced, opens up the possibility of further studies to identify the genetic basis of sleep regulation.

such that they are up-regulated after sleep deprivation and in short-sleeping mutants. We conclude that homeostatic sleep drive promotes sleep at least in part by increasing expression of RYE.

## Results

### Identification of redeye, a short-sleeping mutant

As previous screens have identified sleep components on the X chromosome and the second chromosome (*Cirelli et al., 2005*; *Koh et al., 2008*; *Stavropoulos and Young, 2011*), we sought to identify sleep-altering mutations on the third chromosome in *Drosophila*. To this end, we generated a stock of iso31 (*Ryder et al., 2004*) flies carrying a newly isogenized third chromosome and treated males of this stock with 10–25 mM ethyl-methanesulfonate (EMS). Individual male progeny were bred to achieve homozygosity of the third chromosome, and females of the F3 generation were tested for daily sleep patterns in the presence of light:dark cycles (*Koh et al., 2008*). As seen in previous screens (*Koh et al., 2008*), sleep amounts were normally distributed in the 1857 lines assayed (*Figure 1A*, *Figure 1—source data 1*). Two homozygous mutant lines showed a severe reduction of sleep length, and we focused on one of these that we named *redeye* (*rye*). *rye* homozygotes show a >50% reduction in both daytime and night-time sleep (*Figure 1C*). The activity (average number of beam crossings during periods of activity) of *rye* mutants is comparable with that of wild type controls, suggesting it is not a hyperactive mutant (*Figure 1B*). *rye* heterozygotes have slightly less sleep than wild type, suggesting that the *rye* mutation is partially dominant (*Figure 1B,C*). The dramatic reduction of baseline sleep results largely from a shortening of the average sleep episode duration (*Figure 1D*), which is indicative of a defect in sleep maintenance. The average number of sleep episodes at night is actually increased significantly in *rye* mutants, perhaps because the sleep homeostat senses sleep loss and compensates by initiating more bouts. We also assayed *rye* mutants in sleep deprivation assays and found that they did not lose much additional sleep in response to mechanical stimulation (data not shown). We surmise that an inability to sustain sleep bouts leads to the buildup of high sleep need in *rye* mutants, which makes them resist any further sleep loss.

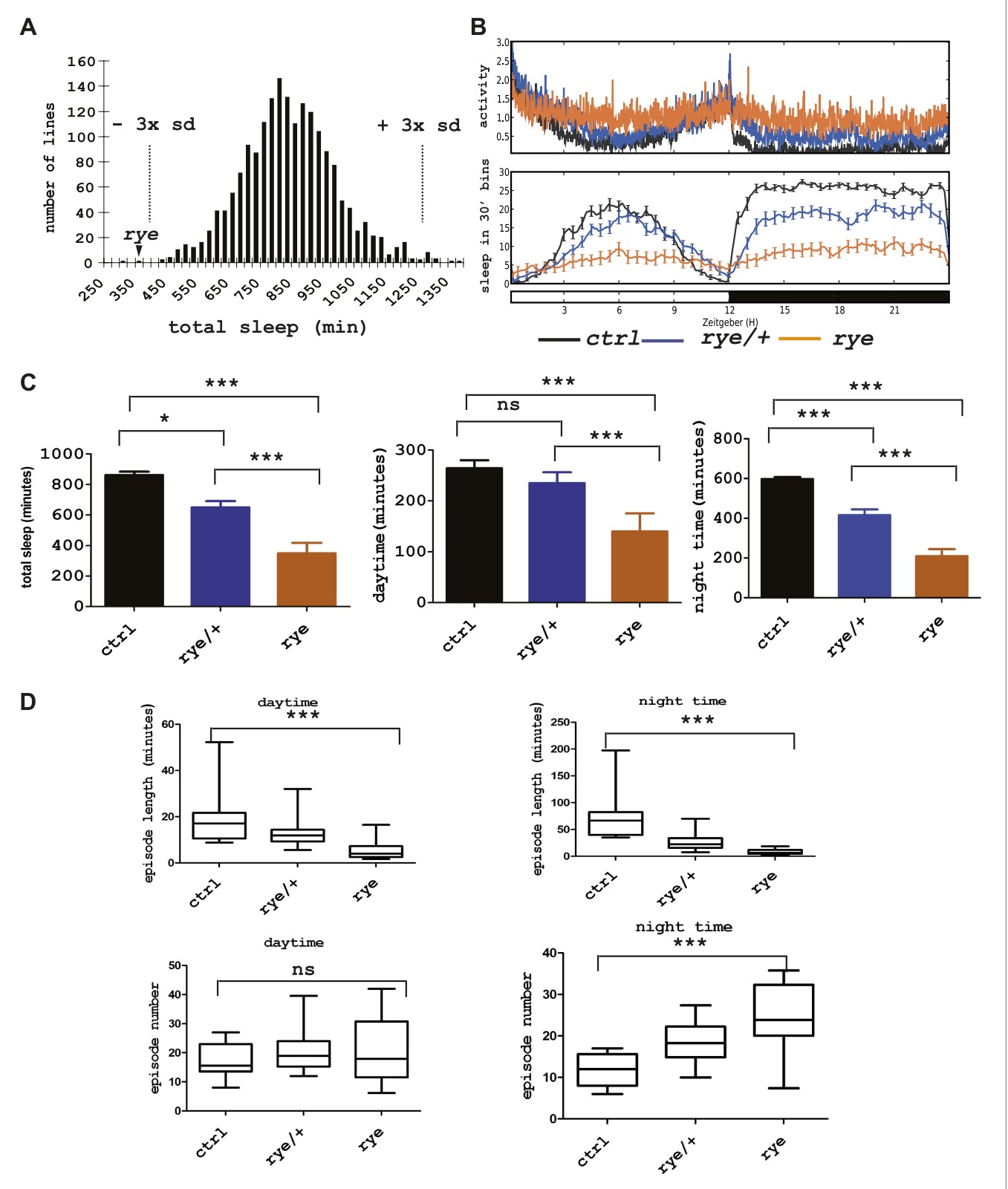

**Figure 1**. Identification of a short sleep mutant, *redeye* (*rye*), through a chemical mutagenesis (EMS) screen. (**A**) Histogram depicting average sleep levels in females from homozygous EMS-mutagenized lines (n = 1857). The mean sleep value for each mutant was calculated by assaying sleep in 4–8 individual flies in the presence of 12 hr L-12 hr D cycles. Average sleep for all lines is indicated on the X axis in increments of 25 min. The Y axis depicts the number of mutant

*Figure 1. Continued on next page*

*Figure 1. Continued*

lines within each group. The dashed lines mark the sleep values that correspond to plus/minus 3× standard deviation of the mean. The arrowhead indicates the *redeye* mutant. (**B**) Top: average activity pattern of females from control lines and from *rye* mutants recorded in 12hr L-D cycles (n = 14–16 in 5 days). Bottom: sleep profiles with standard error bars. Black: lines isogenized on the 3ʳᵈ chromosome for chemical mutagenesis; Blue: flies heterozygous for *rye* and the isogenized control chromosome; Orange: *rye* homozygotes. (**C**) Mean values of total sleep, daytime sleep and night-time sleep for *rye* mutants (*Figure 1—source data 1*). For each genotype 14–16 flies were assayed over a 5 day period. Bars represent standard error. One-way ANOVA was performed followed by Tukey post hoc analysis. * represents p<0.05, **p<0.01 and ***p<0.001. (**D**) Sleep quality of *rye* mutants: Daytime and night-time sleep episode length and episode number are plotted in the box-and-whisker diagram (*Figure 1—source data 1*). The middle line represents the median value; Bottom and top line of each box represent 25% and 75% respectively; Bottom bar and top bar represent 5% and 95% respectively. ns: not significant, ***p<0.001.

The following source data are available for figure 1:

**Source data 1**. Measurement of sleep duration for all EMS mutants and sleep analysis of *rye* mutants.

## Normal circadian rhythms in rye homozygotes

As the circadian clock regulates the timing of sleep and some clock mutants show reduced sleep (*Hendricks et al., 2003*), we also assayed *rye* flies for defects in circadian rhythms. These endogenously generated rhythms are best monitored under constant conditions, in the absence of cyclic environmental cues, so we monitored behavior in constant darkness (DD). *rye* mutants display robust rest:activity rhythms in DD (~62% rhythmicity) despite showing prolonged duration of activity (*Figure 2A*; *Table 1*). Of the 'arrhythmic' *rye* homozygotes detected, half (6/11) actually displayed a rhythm of ~12 hr period length, which likely resulted from persistence of bimodal behavior (typically seen in light:dark) in DD. Thus, most *rye* homozygotes are rhythmic. Consistent with the intact circadian behavior, circadian cycling of the clock protein, PERIOD (PER), is normal in central clock cells of *rye* mutants. The peaks and troughs of expression, as well as the subcellular localization, at different times of day are similar to those in *iso* controls (*Figure 2B*). Thus, PER is largely nuclear at ZT20, entirely nuclear at ZT2, expressed at low levels at ZT8 and undetectable at ZT14. We conclude that only the homeostatic regulation of sleep is affected in *rye* mutants.

## Identification of rye as an α subunit of the nicotinic acetylcholine receptor

We identified the sleep-altering lesion in *rye* mutants through a combination of recombination mapping and deep-sequencing. Despite starting with a newly isogenized line, deep-sequencing identified many nucleotide changes in the *rye* background relative to the parental control and so considerable mapping was required to pinpoint the relevant mutation. Meiotic recombination analysis positioned *rye* between two genetic markers, *thread* and *curled,* in the centromeric region of the 3ʳᵈ chromosome. Further mapping studies, using self identified single nucleotide polymorphism (SNP) markers, narrowed it down to ~11 mega-bases (*Figure 3A*, *Figure 3—source data 1*). For finer localization, we utilized the genome-wide sequence data, which provided 40× and

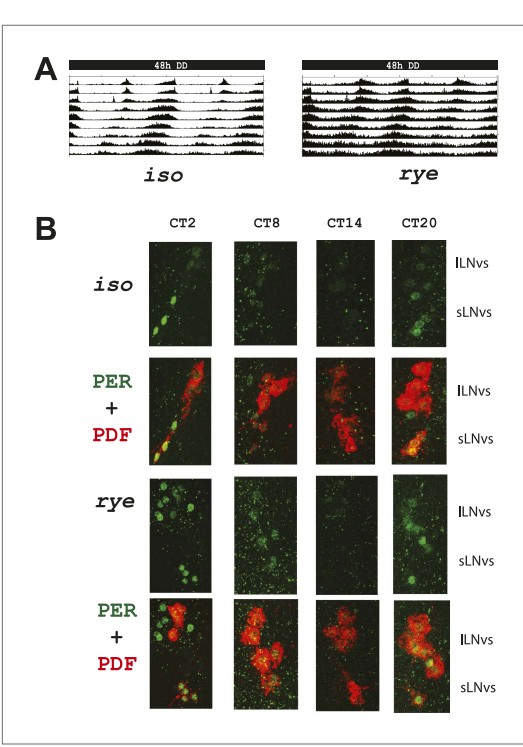

**Figure 2**. *rye* homozygotes display normal circadian rhythms and clock protein cycling. (**A**) Double plotted activity records of *iso* controls and *rye* homozygotes in DD. Rhythmic, but prolonged cross-beam activity was recorded in *rye* homozygotes. (**B**) Immunostaining of PERIOD in ventral lateral neurons (LNvs) of *iso* controls and *rye* homozygotes at CT time points on the second day of DD following LD entrainment. PER oscillates in these clock neurons in *iso* as well as in *rye* homozygotes. PDF labels the small and large LNvs.

**Table 1.** Rhythmic behavior of rye homozygotes in constant darkness

| Genotype | Rhythmicity% (n)* | Period ± SEM (h)† | FFT ± SEM† |
|---|---|---|---|
| W[1118] (iso) | 93.8% (30/32) | 23.21 ± 0.04 | 0.051 ± 0.005 |
| rye | 62.1% (18/29) | 23.66 ± 0.09 | 0.054 ± 0.005 |

*Flies were entrained to a light–dark cycle (12 hr/12 hr) for 3 days before being moved into constant darkness (DD). Behavior was analyzed from day 3 to day 11 in DD, and flies with a fast fourier transform value (FFT) above 0.01 were considered rhythmic.
†Period lengths and FFT values of rhythmic flies are listed as average value plus minus the standard error of mean (SEM).

10× genomes worth of nucleotide sequence, corresponding to 95% and 88% coverage, for the *rye* and *iso31* genomes respectively. Following alignment with the published *Drosophila* genome, unique nucleotide polymorphisms present in *rye* mutants were pooled together as potential EMS-induced mutations (n = 26,224). Within the 11 Mb region identified through recombination mapping, there were 1457 potential mutations, but only nine that produced an amino-acid change in coding regions. Sanger sequencing allowed us to exclude seven of these. Two turned out to be deep-sequencing errors and five polymorphisms were also present in the wild type *iso* stock but not detected due to the low coverage of deep-sequencing in this stock. One polymorphism, in a gene annotated CG7320, was at a site that was also polymorphic in *iso*, although the amino acid change was different in the two strains. Because of the change in *iso*, which does not produce a sleep phenotype, we considered CG7320 a less likely candidate for *rye*. The last of the nine changes identified by deep-sequencing was a C-T transition (typical for EMS-induced mutations) that leads to a threonine to methionine change in CG12414 and is present only in *rye* mutants (*Figure 3B*). CG12414 encodes an α subunit of a nicotinic acetylcholine receptor (nAchR) (*Lansdell and Millar, 2000*). Five alternatively spliced variants produced by this gene are predicted to generate three possible open reading frames with one (pC/pG) containing an intact ligand bind domain (LBD) (*Figure 3C*). The predicted protein is homologous to several nAChR subunits, including α3, α2, α6, α4 and α7 in order of similarity. Alignment of CG12414 with α3 and α7 using the ClustalW2 algorithm shows that the *rye* mutation is at the junction between the LBD and the trans-membrane domain (TM) (*Figure 3D,E*). Interestingly, this junction region is highly conserved across species and the specific threonine mutated in *rye* is conserved in all α7 subunits (*Figure 3D*).

## Functional analysis of rye

The genetic mapping and deep-sequencing data combined implicated CG12414 as a candidate gene responsible for the *rye* sleep phenotype. To verify this function for CG12414, we attempted to rescue *rye* mutants with a CG12414 transgene. To drive expression in areas that normally express this gene, we cloned the CG12414 promoter region (1.8 kb) and generated a GAL4 construct. This GAL4 is expressed broadly in the *Drosophila* brain (*Figure 4A*). We also cloned the CG12414 cDNA into a UAS construct and crossed the GAL4 and UAS transgenes into an *iso31* stock to ensure a homogenous genetic background. These transgenes were then introduced into *rye* mutants. While either transgene alone did not alter the *rye* short sleep phenotype, the *rye* promoter driving RYE expression (*ryeP > uas-rye*) partially rescued the mutant phenotype and restored total sleep to levels seen in *rye* heterozygotes (*Figure 4B*). This is expected, given that *rye*[T227M] is partially dominant over wild type. In a wild type background, *ryeP > uas-rye* does not increase sleep length (data not shown), suggesting that the increase in sleep is specific to the mutant. Thus, we identified the gene responsible for the short sleep phenotype in *rye* mutants, and henceforth refer to this gene CG12414, previously called nAChR-80B, as *redeye*.

To further characterize *rye* function, we knocked down RYE expression through RNA interference. A *rye* RNAi line (#11392) from the VDRC stock center in Vienna was crossed into the *ryeP*-GAL4 line along with a UAS-Dicer2 transgene, included to improve efficacy of the RNAi. Levels of RYE were reduced (*Figure 4D*), as was the total sleep length (*Figure 4C*). Since *rye*[T227M] reduces sleep as well (*Figure 1*), these data suggest that the T227M mutant allele causes loss of RYE function. The sleep phenotype produced by *rye* knock-down is rather mild in comparison with the phenotype of the homozygous *rye* mutant, possibly due to inefficient knockdown in relevant cells or because *rye*[T227M] is a dominant-negative mutation. The nicotinic acetylcholine receptor forms a pentameric structure,

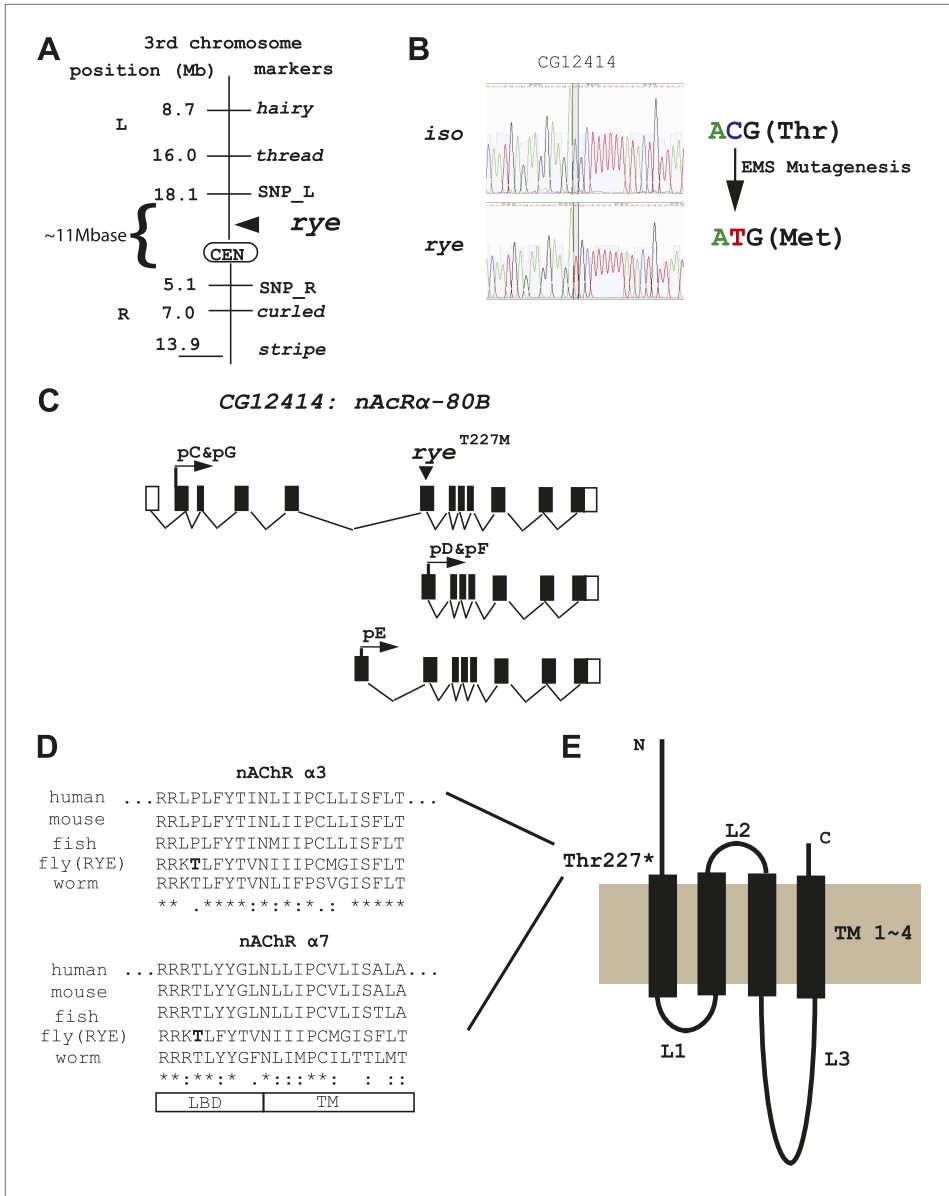

**Figure 3**. Genetic mapping and genome-wide deep-sequencing reveal a missense mutation in a nicotinic acetylcholine receptor α subunit gene in *rye* mutants. (**A**) Through genetic mapping, using classical phenotypic markers and newly identified SNP markers, *rye* was mapped between two SNP markers (SNP_L and SNP_R) near the centromere on the 3rd chromosome. (**B**) Paired-end genomic deep-sequencing of *rye* mutants identified a missense mutation in CG12414. We confirmed this through Sanger sequencing. This C-T/G-A transition that generated the *rye*[T227M] allele is typical of EMS-induced mutations. (**C**) Schematic representation of the *rye* candidate gene, a nicotinic acetylcholine receptor (CG12414: nAcRα-80B). The gene spans ~100 kb with alternative splicing predicted to produce five transcripts (RC-RG). The spliced forms are predicted to translate into three proteins isoforms (pC-pG). pC(pG) produces the largest protein with the longest ligand binding domain (LBD). (**D**) Alignment of the partial sequence of *Drosophila* RYE with nAChR α3 and α7 subunits in other animals using ClustalW2. The region shown is at the boundary of the ligand binding domain (LBD) and transmembrane domain (TM), and is evolutionarily conserved. T227 in RYE is marked in bold form. '*' identity; ':' high similarity; '.' similarity. (**E**) Protein sequence analysis predicts four transmembrane domains (TM) in RYE, which is typical of most nAChR proteins. RYE appears to contain a single ligand binding domain (LBD) in the extracellular region, and four TMs with three loop regions (L1-L3). The mutated threonine227 is in the LBD, close to the beginning of the TM.

The following source data are available for figure 3:

**Source data 1**. Recombination mapping of the *rye* mutation using a chromosome marked with visible markers *h,th,cu,sr,e,* and using SNP markers.

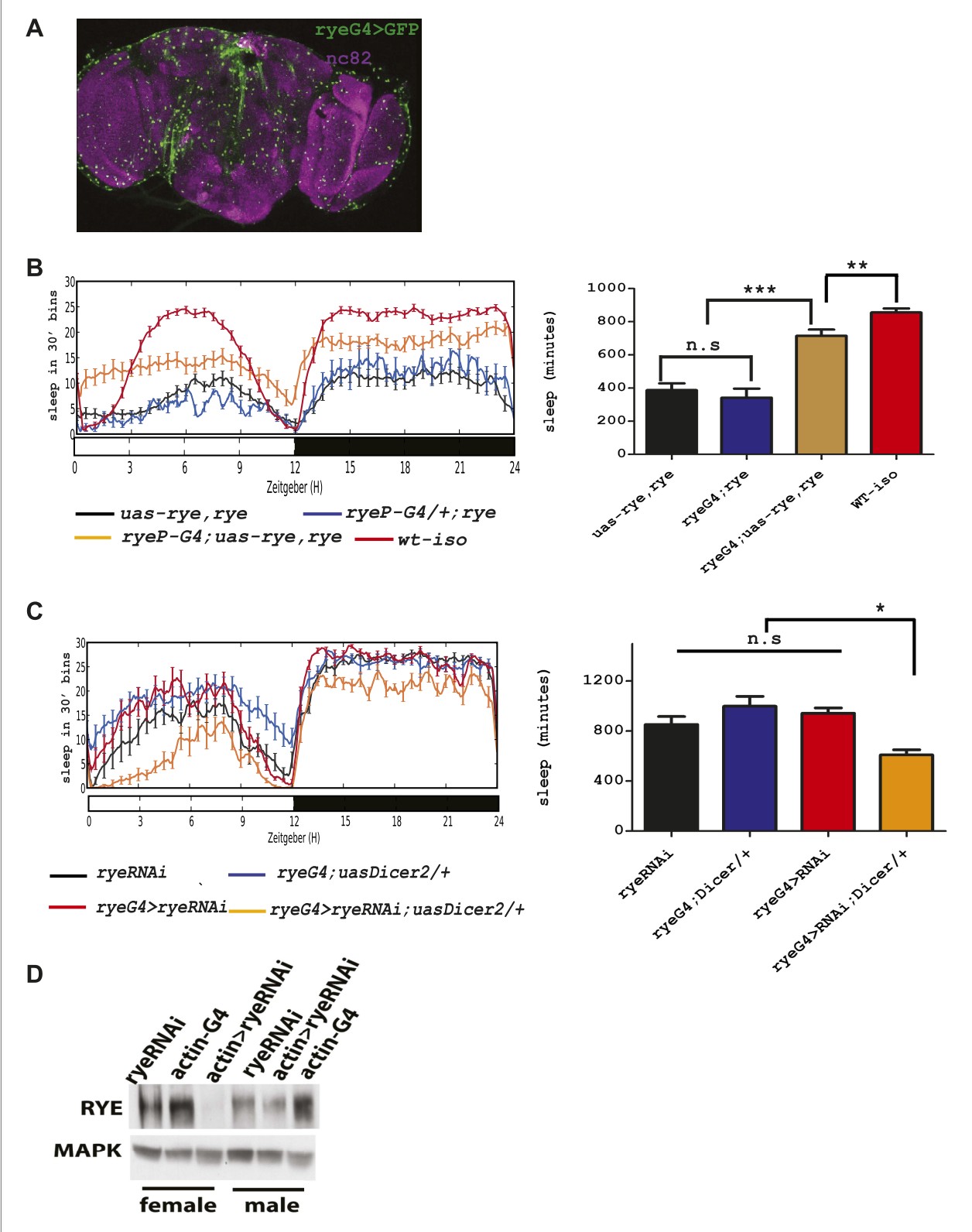

**Figure 4**. An alpha subunit of the nicotinic acetylcholine receptor accounts for the *rye* mutant phenotype. (**A**) Expression pattern of ryeP-GAL4. Rye-GAL4 was used to express nGFP, which is visualized with an anti-GFP antibody. The anti-nc82 staining marks the neuropil. (**B**) Expression of the alpha subunit, as described in **Figure 3**, increases sleep duration in *rye* mutants (**Figure 4—source data 1**). Left: sleep profiles, with standard error, of *rye* mutants and mutants expressing a UAS construct of the putative *rye* cDNA under control of its own promoter (*ryeP-Gal4)* in a 12:12 LD cycle. Right: quantification
*Figure 4. Continued on next page*

*Figure 4. Continued*

of sleep length. **p<0.01, ***p<0.001. (**C**) Reduction of *rye* expression in *rye* neurons through the expression of an RNAi construct, together with Dicer2, diminishes sleep length (*Figure 4—source data 1*). Left: sleep profile in a 12:12 LD cycle. Right: quantification of sleep length. *p<0.05. (**D**) Western blot analysis shows reduced expression of RYE when actin-GAL4 is used to drive *rye* RNAi (VDRC#11392) with Uas-Dicer2 in female and male flies.

The following source data are available for figure 4:

**Source data 1**. Sleep behaviour of transgenically rescued *rye* mutants and *rye* RNAi lines.

consisting of five homo-oligomeric or hetero-oligomeric subunits (*Miwa et al., 2011*), so a dominant negative mutation in any one subunit may well interfere with activity of the complex.

## RYE interacts with SSS

We next addressed if *rye* interacts with other known sleep mutants, in particular with a mutant we identified previously, *sleepless (sss)*, that has an extreme short-sleeping phenotype (*Koh et al., 2008*).

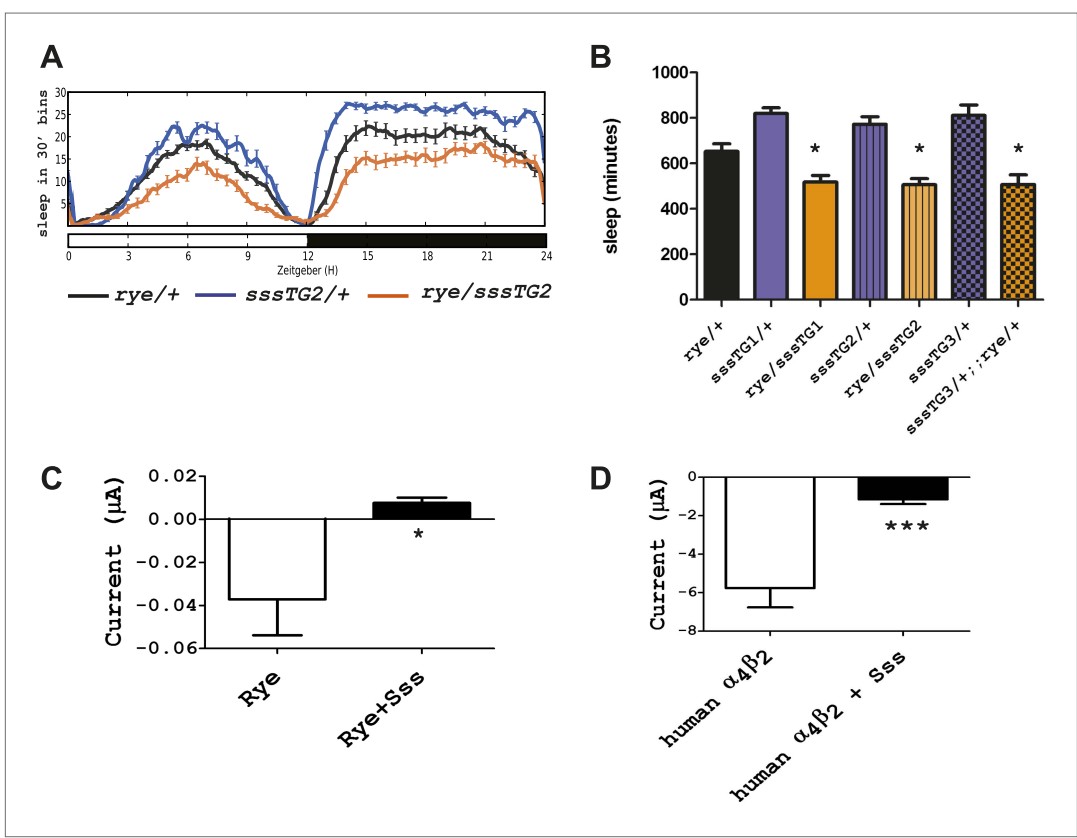

**Figure 5**. Overexpression of *sss* exacerbates the *rye* phenotype by repressing RYE activity. (**A**) An extra copy of SSS reduces sleep in *rye* heterozygotes. Sleep profile of *rye* heterozygotes with or without a genomic *sss* transgenic line inserted on the third chromosome (*sssTG2*) and of *sss* transgenics alone. (**B**) Quantification of sleep in the genotypes as indicated (*Figure 5—source data 1*). TG1 and TG3 are additional independent genomic *sss* transgenes inserted on the 3rd and X chromosomes respectively. Sleep reduction was observed in *rye* transheterozygotes with all three *sss* transgenic lines. *p<0.05. (**C**) Heterologous expression of SSS reduces current following ACh application in *Xenopus* oocytes expressing *Drosophila* RYE and human nAChR β2 (*Figure 5—source data 1*). *p<0.05. (**D**) Heterologous expression of SSS reduces current following ACh application in *Xenopus* oocytes expressing human nAChR α4β2 (*Figure 5—source data 1*). ***p<0.001.

The following source data are available for figure 5:

**Source data 1**. Sleep analysis of *rye* mutants overexpressing sss, and whole-cell membrane current recording of *Xenopus* oocytes.

While transheterozygotes of *rye* and *sss* (*sss/+*; *rye/+*) showed sleep duration comparable to that of *rye/+* alone (data not shown), *rye* heterozygotes carrying a genomic *sss* transgene, which increases SSS expression above that of wild type controls, showed a further reduction of sleep (*Figure 5A*). The effect was observed with three independent insertions of the *sss* transgene, indicating that it was not due to insertion site effects (*Figure 5B*). We note that this *sss* transgene does not have a significant effect on sleep in wild type flies although it completely rescues the sleep phenotype of *sss* mutants (*Koh et al., 2008*). Thus, the effect is specific to *rye* mutants.

SSS is a GPI-anchored protein that regulates the Shaker potassium channel and its predicted structure resembles that of a specific class of toxins (*Wu et al., 2010*; *Dean et al., 2011*). It is most similar to the mammalian Lynx-1 protein, which is known to inhibit activity of the nicotinic acetylcholine receptor (*Miwa et al., 2011*). The fact that SSS exacerbated the phenotype of flies heterozygous for *rye*, which have reduced activity of nAChR, suggested that SSS also regulates nAChRs. We tested a possible Lynx-1-like function of SSS in regulating nAChR activity by using a heterologous expression system, specifically *Xenopus laveis* oocytes. For functional expression of wild type RYE, we expressed it with a human β2 subunit, as insect β subunits typically do not work in heterologous systems (*Millar and Lansdell, 2010*). cRNAs encoding *sss*, wild type *rye* and human β2 were injected into oocytes, and 4–7 days later whole cell currents were measured following application of ACh (*Kuryatov and Lindstrom, 2011*). Heterologous expression of insect nAChRs has, in general, been challenging (*Millar, 2009*), which accounts for the low current values observed (*Figure 5C*). However, we still detected a significant reduction of nAChR current upon co-expression with SSS (*Figure 5C*). We also co-expressed in vitro transcribed *sss* and human α4 and β2 subunits and found that SSS also inhibits the robust current produced by the human receptor complex (*Figure 5D*). Thus, in addition to its sleep-promoting role reported previously, SSS may also promote wake by repressing activity of nAChRs. As discussed below, loss of the sleep-promoting role has a dominant effect in *sss* mutants.

## RYE is expressed cyclically in association with the sleep state

We next asked if levels of *rye* vary over the course of the day. Groups of wild type flies were housed in a 12 hr-light and 12 hr-dark incubator and harvested at different Zeitgeber times (ZT). Zeitgeber time defines time based on the environmental stimulus, so ZT0 corresponds to light-on and ZT12 to light-off. RNA was extracted and subjected to quantitative PCR analyses. While *period* (*per*), a well-known circadian clock component, oscillates with a circadian rhythm (*Zheng and Sehgal, 2012*), *rye* mRNA levels were found to be constant (*Figure 6A*).

To examine protein levels of RYE, we generated an antibody against the cytoplasmic L3 domain of RYE (*Figure 3E*), which is the most variable region across all nAChR subunits (*Lindstrom, 2003*). On western blots, the antibody recognizes a band (or sometimes a doublet) of ~60 Kd, which is the size predicted for the full-length isoform. Surprisingly, in contrast to the flat mRNA levels, levels of RYE protein show a robust rhythm with two peaks each day, one at around ZT6-10 and the other at ZT18-22 (*Figure 6B*, *Figure 6—figure supplement 1*). Interestingly, these correspond to the two daily times of sleep, the afternoon siesta and night-time sleep (*Figure 1B*), suggesting that RYE is expressed in phase with sleep. The cycling of RYE persists in constant darkness, as do rhythms of sleep (data not shown). However, RYE does not appear to be regulated like other cycling clock components, such as PER, which usually show only one peak (*Figure 6B*). Furthermore, we found that RYE continues to cycle in *Clk^{irk}* flies that lack a circadian clock, although the phase of RYE expression tends to be more variable in these flies (*Figure 6C*, *Figure 6—figure supplement 2*). These data suggest that the RYE oscillation per se does not require process C of the two process model, but may depend upon process S (sleep homeostasis). However, the circadian clock may function to provide more precise timing of RYE expression, perhaps through its control of sleep onset and offset.

## Expression of RYE is under control of the sleep homeostat

Increases in RYE expression could occur as a consequence of sleep, and thereby reflect the sleep state, but in that case levels should remain high throughout the sleep state. Given that RYE levels do not remain high throughout the night or through the afternoon siesta, another possibility is that elevations in RYE correspond to sleep drive and so occur at the time of sleep onset. To test this idea, we assayed RYE expression in short-sleeping mutants. These mutants have low sleep levels, but probably have high sleep drive that cannot be implemented. As shown in *Figure 7*, for the most part RYE expression exhibits two peaks in these mutants, but the phase is more variable (*Figure 7—figure supplements 1–3*),

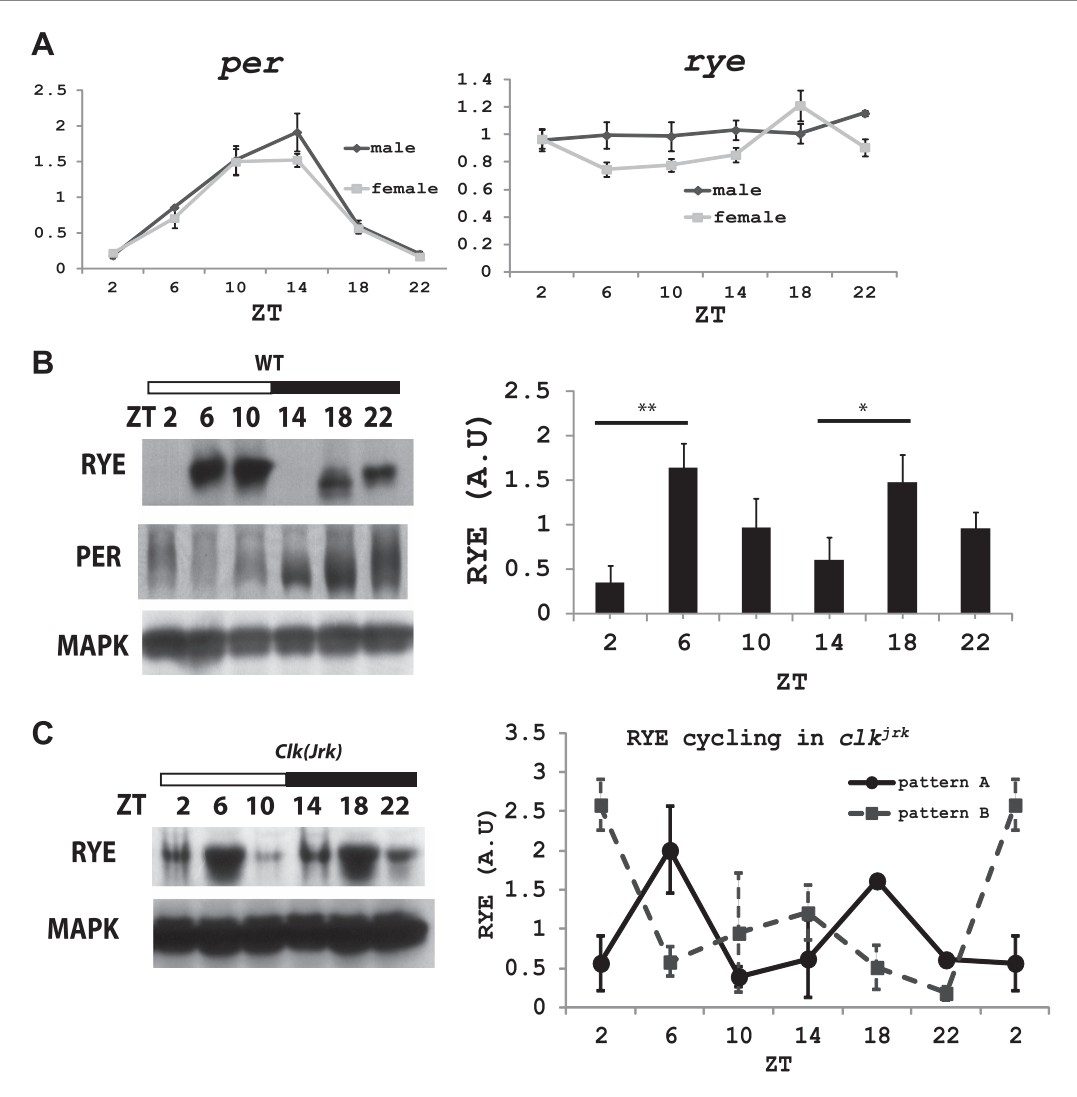

**Figure 6**. The RYE protein is expressed cyclically in association with the sleep state, but independently of the circadian clock. (**A**) Quantitative PCR analyses show oscillations of *per* mRNA (left panel), but constant levels of *rye* mRNA (right panel). *actin* was used as a normalization control (***Figure 6—source data 1***). (**B**) Left: a representative western blot of head extracts from wild type flies shows cyclic expression of RYE with two daily peaks, in the middle of the day (ZT6-10) and in the middle of the night (ZT18-22), corresponding to the sleep state (***Figure 1B***). PER, in contrast, shows only one daily peak. MAPK was used to control for loading. Right: densitometry quantification of western blots with error bars representing standard error (n = 8). RYE value at ZT0 is set as 1 (***Figure 6—source data 1***). *p<0.05, **p<0.01. (**C**) RYE cycling is similar to wild type in *Clk^{jrk}* flies, indicating that cycling per se does not require a functional clock. However, the phase is variable, so for quantification purposes five independent experiments were split into two groups of roughly similar phase (***Figure 6—source data 1***).

The following source data and figure supplements are available for figure 6:

**Source data 1**. qPCR analysis of *per* and *rye* expression during LD cycle and densitometry quantification of RYE expression.

**Figure supplement 1**. RYE expression in a LD cycle (repeats).

**Figure supplement 2**. Supporting data for ***Figure 6C***.

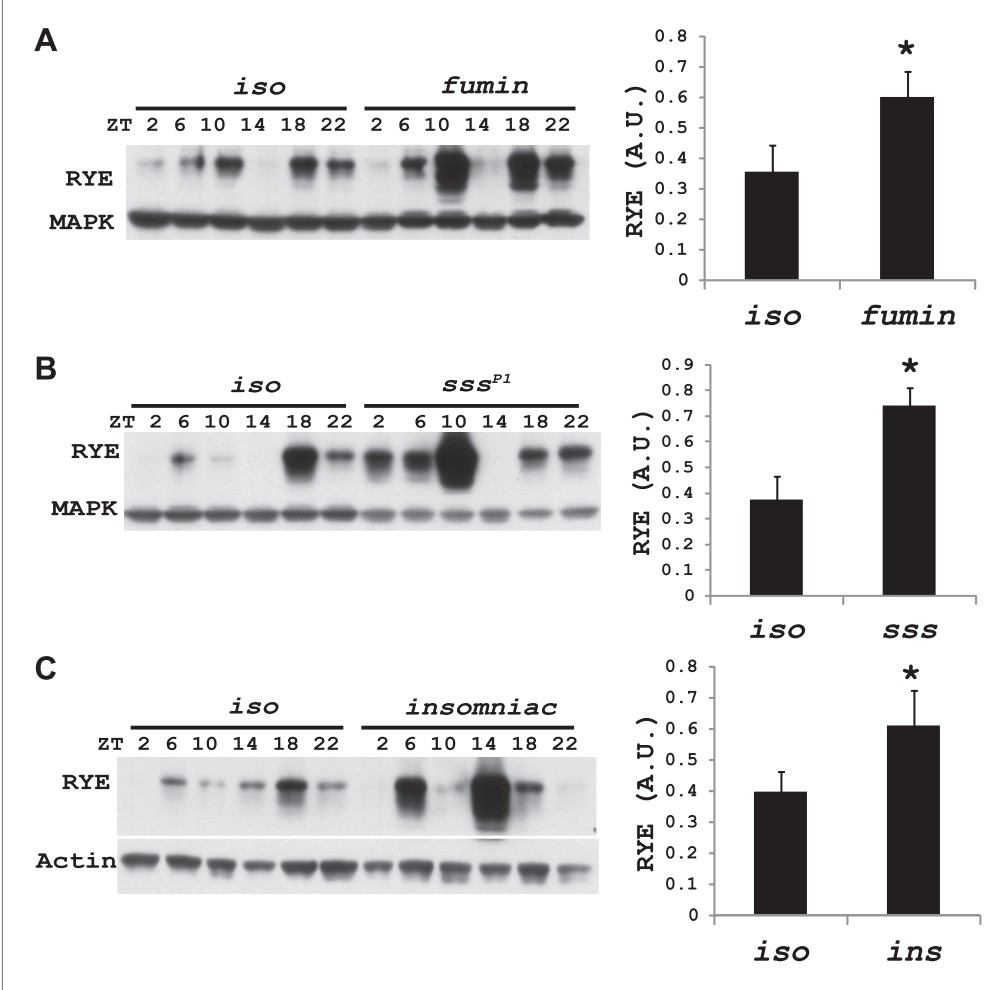

**Figure 7**. RYE levels are elevated in short-sleeping mutants. (**A**) Western analyses of RYE expression over the course of a day in *iso* controls and *fumin* mutants. While RYE still cycles in *fumin*, the mean value of all six daily time points from three independent experiments (n = 3) shows that overall levels of RYE are higher than in wild type controls (*Figure 7—source data 1*). *p<0.05. (**B**) Western analyses of RYE expression over the course of a day in *iso* controls and *sss* mutants. As in **A**, the quantification reflects the mean across the day from three independent experiments (n = 3). *p<0.05. (**C**) Western analyses of RYE expression over the course of a day in *iso* controls and *insomniac* mutants. Quantification as above (n = 3). *p<0.05.

The following source data and figure supplements are available for figure 7:

**Source data 1**. Densitometry quantification of RYE expression in short sleep mutants.

**Figure supplement 1**. Supporting data for *Figure 7A*.

**Figure supplement 2**. Supporting data for *Figure 7B*.

**Figure supplement 3**. Supporting data for *Figure 7C*.

---

indicating that defects in sleep behavior affect the RYE expression pattern. Interestingly, overall RYE values are higher in *fumin* (**Kume et al., 2005**; *Ueno et al., 2012*), *sss* (**Koh et al., 2008**) and *insomniac* (**Stavropoulos and Young, 2011**) mutants than in wild type controls. As noted above, the *sss* mutation affects channel activity, *fumin* is a mutation in the dopamine transporter (short sleep is thought to result from the high levels of dopamine in the synaptic cleft) and *insomniac* is thought to affect protein turnover. The one feature these mutants have in common is short sleep, yet they all have elevated

RYE expression during an LD cycle (*Figure 7A–C*). Based upon these data, we suggest that RYE reflects sleep need. It is elevated at the time of sleep onset and is further increased in short-sleeping mutants because they have high sleep need.

If the expression of RYE changes in response to the accumulation of sleep need, it should also be affected by acute sleep deprivation. We mechanically deprived flies of sleep in the second half night (6 hr) in an LD cycle, and confirmed, through the presence of a substantial sleep rebound the following morning, that the flies were successfully deprived (*Figure 8A*; *Koh et al., 2008*). RYE levels were assayed during the 5 hr sleep rebound window. In the non-deprived control, RYE levels were low at ZT0 and high at ZT5, as noted above. Sleep-deprived flies had high levels of RYE immediately following deprivation (ZT0), but levels were low at ZT5 (*Figure 8A*, *Figure 8—figure supplement 1*). This fits the profile for sleep drive, which is expected to be high at the end of deprivation, but then dissipated over the course of rebound sleep.

## Discussion

The molecular mechanism of sleep homeostasis is a mystery and a subject of intense research in the sleep field (*Suzuki et al., 2013*). In addition to the investigation of mechanisms underlying sleep drive, considerable effort is being put into identifying biomarkers of sleep need. Based on what we know about the so-called sleep homeostat, which consists of increasing sleep pressure during wakefulness and dissipation of such pressure following sleep, we suggest that a component or direct output of the homeostat should satisfy three criteria: (i) the gene product should regulate the sleep:wake cycle (i.e., genetic alleles of this gene should have some sleep phenotype); (ii) expression levels or activity of the gene product should go up during wakefulness or during sleep deprivation and (iii) expression levels or activity should decrease after sleep. The function of RYE and the molecular kinetics of the RYE protein largely satisfy these criteria. However, while RYE builds up during sleep deprivation, it does not accumulate gradually over the wake period in a daily cycle. Rather, it displays a marked increase close to the time of sleep onset, suggesting that it is not a central component of the homeostat, but responds to an upstream homeostatic signal, perhaps when that signal reaches a certain threshold. The fact that over-expression of RYE does not promote sleep also supports the idea that it is not the sleep-inducing homeostatic signal (further discussed below). Nevertheless, RYE is not simply a sleep output gene or sleep biomarker. It is required for implementation of signals from the homeostat and it functions to maintain sleep. Thus, we propose that *rye* is a sleep-regulating gene immediately downstream of the homeostat.

### RYE expression reflects sleep need

We suggest that RYE represents a molecular correlate of delta power, a characteristic of an electroencephalogram (EEG) that reflects sleep drive. Recently, a few other molecules were reported to change with sleep drive, but the effects were at the level of the mRNA, the magnitude of the increase was less than we report here for RYE and loss of the molecules did not affect baseline sleep duration (*Seugnet et al., 2006*; *Maret et al., 2007*; *Naidoo et al., 2007*). In addition, only one is expressed cyclically (*Nelson et al., 2004*), indicating that others reflect sleep drive only under pathological conditions of sleep deprivation. RYE levels oscillate robustly in a daily cycle, although the phase is not as coherent as seen for circadian clock proteins. The timing of the peak varies within a temporal range, such that there is almost always a daytime peak and a night-time peak but not necessarily at the exact same time (*Figure 6—figure supplement 1*). We suggest that RYE cycles under control of the sleep homeostat, which may not time behavior as precisely as the circadian clock, perhaps because sleep can be influenced by many factors. The variability in RYE cycling is particularly pronounced in short-sleeping mutants and in the $Clk^{Jrk}$ circadian clock mutant (*Figure 6—figure supplement 2*, *Figure 7—figure supplements 1–3*), suggesting that the clock does influence RYE expression although it is not required for its cycling in an LD cycle. Interestingly, RYE cycles exclusively at the level of the protein, indicating translational or post-translational mechanisms. It is worth noting that a recently identified sleep regulator, Insomniac, is a component of specific protein degradation pathways in the cell (*Stavropoulos and Young, 2011*). Although our study indicates that RYE cycling does not require Insomniac (*Figure 7C*), it is possible that it is regulated by other protein turn-over machinery. Thus, translational/posttranslational regulation appears to be part of the mechanism of sleep homeostasis.

### RYE is required to maintain the sleep state

We show that RYE not only reflects sleep drive, but is also required for sleep maintenance (*Figures 1, 4 and 8*). Given that RYE is induced by sleep deprivation and it promotes sleep, one might expect

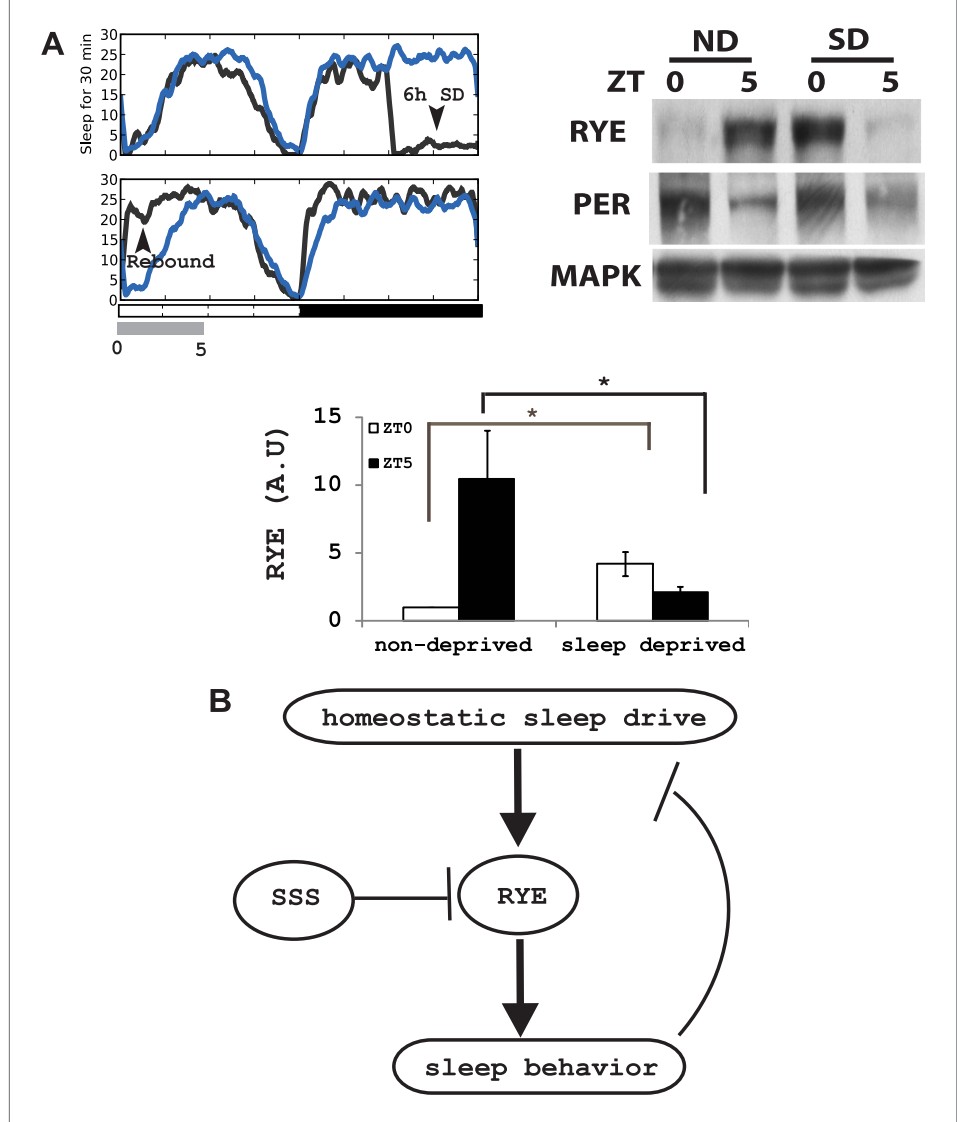

Figure 8. RYE expression is under control of the sleep homeostat. (A) Left: sleep profile of sleep deprived flies. Blue line: non-deprived controls, Grey line: sleep deprived flies. Arrowheads point to the 6 hr sleep deprivation (SD) window (ZT18-24) and sleep rebound the following morning (ZT0-5). Right: a representative western blot of fly head samples at ZT0 and ZT5 in sleep-deprived (SD) animals and non-deprived (ND) controls. RYE levels are high immediately following sleep deprivation (SD0) and dissipate after sleep rebound (SD5). In contrast, the circadian clock is not affected by SD, since PER cycling is comparable in the SD group and the non-deprived control (ND). Densitometry quantification of the western data (n = 5) is shown below with error bars showing standard error (*Figure 8—source data 1*). *p<0.05. (B) Model for the role of RYE in the homeostatic regulation of sleep: We propose that RYE is required to maintain sleep, and posttranslational regulation of RYE reflects homeostatic sleep drive. Sleep need builds up during wakefulness and upregulates levels of RYE, which are essential to maintain sleep behavior. Homeostatic drive dissipates during sleep, and levels of RYE are reduced, leading to wakefulness. SSS represses RYE activity, thus acting as a wake-promoting factor in this particular context.

The following source data and figure supplements are available for figure 8:

**Source data 1**. Densitometry quantification of RYE expression after SD.

**Figure supplement 1**. Supporting data for *Figure 8A*.

over-expression of the protein to increase sleep. However, transgenic expression of *rye* in a wild type background does not increase sleep, suggesting that while *rye* is necessary, it is not sufficient for sleep onset. We cannot exclude the possibility that RYE functions together with other signals as part of the sleep-inducing homeostatic drive. On the other hand, it is also possible that transgenic expression does not produce adequate amounts of RYE protein in relevant cells. This might be the case if RYE is tightly regulated at the level of protein stability. For the moment, though, we prefer the parsimonious explanation noted above, that RYE is not part of the homeostat, but immediately downstream of it.

Acetylcholine signaling has been long proposed as an arousal factor, as the nAChR complex is a cation channel that normally promotes neuronal activity and ACh is released during wakefulness in mammals (*Platt and Riedel, 2011*). In contrast, our study indicates that at least one nAChR subunit (RYE) promotes sleep in the fly. There are more than 10 paralogs of nAChR subunits in the fly genome. One possibility is that RYE is expressed specifically in sleep promoting neurons, while other subunits of AChRs are in wake-promoting cells. An increase in ACh during wakefulness may contribute to the accumulation of sleep drive and to the increase in RYE (*Lindstrom, 2003*). Alternatively, sleep drive may increase RYE independently of ACh, but in either scenario, RYE then promotes sleep. The precise site of RYE action is currently not known. *rye-gal4* driven GFP is expressed widely in the brain (*Figure 4A*), but we cannot be sure that endogenous RYE is as widespread, as our antibody was not effective in immunohistochemistry experiments, and we find that GAL4 drivers are often quite promiscuous (unpublished observations).

## SSS inhibits RYE activity and promotes arousal in a sensitized background

SSS was previously identified as a sleep promoting factor, essential for maintaining baseline sleep and for homeostatic rebound (*Koh et al., 2008*). An interaction between *rye* and *sss* is therefore not surprising. What is surprising is that overexpression of SSS promotes wakefulness in $rye^{T227M}$ heterozygotes (*Figure 5A,B*). SSS is a GPI-anchored protein that functions as a neuronal modulator. Previous studies indicate that SSS promotes activity of the voltage-gated potassium channel, Shaker (*Wu et al., 2010*). In this study, we report that SSS acts like a brake on nAChR (RYE) activity (*Figure 5C,D*), as does Lynx-1, a SSS-like molecule, in mammals (*Miwa et al., 2011*). Although the data we show for *Drosophila* receptors used only the RYE α subunit, it is likely that SSS also inhibits activity of other *Drosophila* nAChR receptors. As both $sss^{P1}$ (a null mutation) and $sss^{P2}$ (a hypomorphic allele) are short-sleeping mutants (*Koh et al., 2008*), we propose that the overall effect of SSS is to promote sleep. The reduced sleep in *sss* mutants probably results from an increase of neuronal excitability, through inactivation of potassium channels (Shaker) (*Dean et al., 2011*), or from hyperactivity of nAChR channels in wake-promoting neurons. Thus, typically the sleep-inhibiting effect of SSS, mediated through RYE, is masked by these other more dominant influences. However, in a sensitized background (i.e., *rye/+*), this effect is evident. RYE promotes sleep, and so loss of RYE results in a decrease in sleep, which is further impacted by SSS overexpression (*Figure 8B*).

We note that there are some caveats to these data. For instance, the $rye^{T227M}$ allele could confer a neomorphic function that accounts for the interaction with *sss*. Likewise, the effects in oocytes could be non-physiological, not necessarily reflecting what happens in the fly brain. However, given that we observe interactions in these two very different types of assays, and both assays indicate repression of nAChR function by SSS, which is the effect predicted from the role of the mammalian SSS-like protein, Lynx1, we believe SSS does indeed regulate nAChRs such as RYE. Interactions between SSS and nicotinic acetylcholine receptors are also reported by recent work from another laboratory (W Joiner, personal communication).

It is interesting that genes identified through independent genetic screens in *Drosophila* are turning out to interact with one another. SSS and Shaker were isolated independently as sleep-regulating genes (*Cirelli et al., 2005*; *Koh et al., 2008*), and subsequently shown to interact, and now we find that RYE interacts with SSS. Given that each of these genes represents a relatively infrequent hit in an unbiased screen, the interactions suggest that genetic approaches are converging upon specific sleep-regulating pathways. Interestingly, a recent GWAS study for sleep-altering loci in humans identified significant effects of SNPs in an nAChR subunit as well as in a regulatory subunit of Shaker, suggesting that these mechanisms are also conserved across species (*Allebrandt et al., 2013*).

## Materials and methods

### Generation of an isogenic 3rd chromosome stock

Wild type *iso*[31] (***Ryder et al., 2004***) was crossed with TM2/TM6C, Sb (Bloomington stock #5906), and a single male progeny was selected and crossed with #5906 to balance the 3rd chromosome and generate a line isogenic for this chromosome.

### Behavioral analysis

Flies were housed in Percival incubators (Perry, IA) and beam–break activity was recorded with the Trikinetics DAM system (http://www.trikinetics.com/). Pysolo (***Gilestro and Cirelli, 2009***) software was used to analyze and plot sleep patterns. Sleep deprivation was achieved through repeated mechanical shaking (2 s randomized shaking in every 20-s interval).

### Mapping analysis

A 3rd chromosome marker line *h, th, cu, sr, e* (Bloomington stock #576) was crossed to *rye*/TM6C, Sb, and heterozygote female progeny (*h, th, cu, sr, e/rye*) were further crossed to TM2/TM6C, Sb (#5906). A single recombinant male offspring was back-crossed to #576 to score the genotype of recessive markers and also back-crossed to *rye* for behavioral analysis to score the *rye* genotype. Overlap in the genotypes of recombinants narrowed down the location of the *rye* mutation to the region between markers *th* and *cu*.

Genomic DNA of homozygous recombinants was subject to SNP analysis. The primer set 5′TGTTTAGTGGTGTTGTGTGAGC3′ and 5′GCCGAGTGTCATCGCCTTTG3′ for SNP_L and the primer set 5′AAAGGTCATCTTGCTTCGGAGTTG3′ and 5′GGAGTGGCTTCCTCGTCATC3′ for SNP_R were used for PCR amplification. Nucleotide polymorphisms were identified between *iso* and #576. Thus, those recombinants were scored accordingly and *rye* was further narrowed down to a region between those two SNP markers.

### Sequencing and cloning

Fragmented genomic DNA (~400 bp in size) obtained through the Covaris instrument (Woburn, MA) was subjected to Illumina paired-end DNA library preparation. The libraries were amplified for 10 PCR cycles prior to Illumina Hi-Seq analyses. SNP calling analyses identified nucleotide polymorphisms in both *rye* mutants and *iso* flies.

The primer set 5′GGCTCGATGTGCTTTCAAGAGTTC3′ and 5′CATGCCAGATGAGTGCGTTTC3′ was used to clone the *rye* promoter region from genomic DNA derived from the *iso* stock. The primer set 5′TTTAGGCTTAGTCCGCTACC 3′ and 5′AATGTCGTGGTTTGAAGTGC 3′ was used to clone the full length *rye* cDNA. The PhiC31 integration system was used for injection. The *rye* promoter construct was specifically inserted onto the 2nd chromosome and the Uas-*rye* construct was integrated on the 3rd chromosome.

### Generation of RYE antibodies

The primer set 5′ATGCATCATCACCATCACCATAGTATCTGCGTGACGGTTGTTG3′ and 5′GTGGTGGTGCACAACTGCCAACGTGAATATCC 3′ was used to clone the L3 epitope of RYE. A GST-RYE[L3] fusion protein was expressed in *Escherichia coli*, excised from a PAGE gel and injected into rats for antibody generation.

### Extractions and measurements of DNA, RNA and protein

*Drosophila* genomic DNA: flies (3–15) were frozen and homogenized in DNA extraction buffer. After LiCl/KAc precipitation, supernatants were subject to isopropanol precipitation. DNA pellets were dissolved in TE for sequencing analyses and PCR amplification.

RNA: adult flies were collected at indicated time points and fly heads (10~20) were subject to Trizol extraction (Ambion, Life Technologies, Grand Island, NY). cDNA libraries were made through high-capacity cDNA reverse transcription kits (Applied Biosystems, Life Technologies, Grand Island, NY). Quantitative PCR analysis was performed using SYBR green reagents in the Applied Biosystems 7000 sequence detection system. The primer set 5′AGTTGAATGGAAGCCACCAGC3′ and 5′TGTTCATCCATGTGCCTCAG3′ was used for qPCR analysis of *rye* expression.

Protein: flies were collected at indicated time points and fly heads (~5) were prepared for protein extraction and for Western analyses as previously described (***Luo and Sehgal, 2012***).

## Immunocytochemistry assays

Adult fly brains were dissected in cold PBS buffers at indicated CT time points. Brains were fixed in 4% PFA for 30 m and incubated with 5% Normal Donkey Serum (NDS) for 1 hr at the blocking step. Incubation with primary rabbit anti-PER (1:1000), mouse anti-PDF (1:1000) and rabbit anti-GFP, mouse anti nc82 was performed overnight in the cold room. After extensive washing, brains were incubated with the donkey anti-rabbit or anti-mouse secondary antibody (1:1000) for 2 hr. Images were taken under a Leica confocal microscope.

## Electrophysiological current recording in *Xenopus* oocytes

Oocytes were removed surgically from *Xenopus laevis* and placed in an OR-2 solution containing 82.5 mM NaCl, 2 mM KCl, 1 mM $MgCl_2$, 5 mM HEPES, and pH 7.5. They were defolliculated in this buffer containing 2 mg/ml collagenase type IA (Sigma, St. Louis, MO) for 1.5 hr. After defolliculation, oocytes were incubated at 18°C in 50% L15 (Invitrogen, Carlsbad, CA) 10 mM HEPES, pH 7.5, 10 units/ml penicillin, and 10 g/ml streptomycin at 18°C. 3–6 days after injection, whole-cell membrane currents evoked by acetylcholine were recorded in oocytes at room temperature with a standard two-electrode voltage-clamp amplifier (Oocyte Clamp OC-725; Warner Instrument, Hamden, CT). Recordings were performed at a holding potential of −50 mV. All perfusion solutions contained 0.5M atropine to block responses of endogenous muscarinic AChRs that might be present in oocytes. Acetylcholine was applied by means of a set of 2-mm glass tubes directed to the animal pole of the oocytes. Application was achieved by manual unclamping/clamping of a flexible tube connected to the glass tubes and to reservoirs with the test solutions. The recording chamber was perfused at a flow rate of 15–20 ml/min. cRNAs for human AChR subunits 4 and 2 and *Drosophila sss* were synthesized in vitro using SP6 RNA polymerase (mMESSAGEmMACHINETM, Ambion). Oocytes were injected with 5 ng of cRNA of each of the subunits.

## Acknowledgements

We are grateful to Christine Quake for technical support of this project, Matt Kayser, Dan Cavanaugh, Katarina Moravcevic and Christine Dubowy for critical reading of the manuscript and other members of the laboratory for useful discussions. We thank William Joiner for communicating results prior to publication. We also thank Natania Field for suggesting the name, redeye. AS is an Investigator of the HHMI.

## Additional information

### Funding

| Funder | Author |
| --- | --- |
| Howard Hughes Medical Institute | Amita Sehgal |

The funder had no role in study design, data collection and interpretation, or the decision to submit the work for publication.

### Author contributions

MS, Conception and design, Acquisition of data, Analysis and interpretation of data, Drafting or revising the article; ZY, AK, Acquisition of data, Analysis and interpretation of data; JML, Conception and design, Analysis and interpretation of data; AS, Conception and design, Analysis and interpretation of data, Drafting or revising the article

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
