## [Decision Letter]

Thank you for sending your work entitled “Identification of Redeye, a new sleep-regulating protein whose expression is modulated by sleep amount” for consideration at *eLife*. Your article has been favorably evaluated by a Senior editor and 3 reviewers, one of whom is a member of our Board of Reviewing Editors, and one of whom, Subhabrata Sanyal (Reviewer 3), has agreed to reveal his identity.

The Reviewing editor and the other reviewers discussed their comments before we reached this decision, and the Reviewing editor has assembled the following comments to help you prepare a revised submission.

The consensus of the reviewers was that the identification of *rye* is an important and novel contribution. To our knowledge there are no other protein markers that have been reported that are independent of the clock and external conditions and respond only to sleep drive. RYE levels therefore potentially represent an important new biomarker for sleep deprivation. The reviewers also felt that the data on the interaction of *rye* and *sss* were the weakest part of the paper and that without deeper investigation the precise interpretation of those experiments was difficult. While it would be satisfying to have mechanism, this study is still quite important without a full understanding of the interaction of *rye* and *sss*. We would therefore recommend publication of a suitably revised paper. The full reviews below contain a number of minor things that will help the authors to make the paper stronger, but the main concerns/suggestions are enumerated as follows:

1) Eliminate Figure 5 or cut back on the mechanistic claims and give a much fuller discussion of the caveats surrounding the data (see Reviewer 1 comments).

2) More balanced discussion of the issue of phase and whether *rye* is downstream of S or actually part of the homeostat (see Reviewer 2 comments). The overexpression data suggest the former.

3) Discussion of the issues surrounding the variability of the western blots (see comment from Reviewers 1 and 3).

4) More information on *rye* expression (or GAL4 pattern) and localization of requirement would strengthen the paper, especially if Figure 5 is eliminated. If such data exist it would be good to include them, but we would not require new experiments to be done.

*Reviewer 1*
*comments:*

This is a very interesting study. The authors used the power of genetics to identify a novel sleep regulator: RYE. The fact that RYE levels are apparently controlled by sleep drive and not the clock make this a very novel and important discovery. I believe this may be the first molecule of this class and for that reason I am very enthusiastic about this paper.

Unfortunately, I think that the data that go beyond the characterization of *rye* are not very strong and are in many cases not interpretable in the simple way that the authors try to suggest. I almost wonder if the *sss* connection should not just be taken out of the paper until the authors have more data and can present a believable story. The isolation of the *rye* gene could stand alone as a brief communication. As it is the SSS stuff just muddies the waters. Perhaps if they added more detail about how the mutants are resistant to SD to the basic story?

I am not convinced that there is a specific interaction between SSS and RYE that regulates sleep in WT animals. First, the authors claim that the *sssTg* lines have no effects on sleep, but they show data that the sleep levels in these lines are equivalent to *rye/+*, which has reduced sleep. The fact that now combining them results in even less sleep suggests a simple additive and independent effect of each manipulation on sleep. Second, there are no data on RNAi of RYE + OE of SSS. It is possible that the SSS effect is specific to the mutant allele because it has neomorphic function and that SSS may not regulate WT RYE.

The oocyte data are not deep enough to ameliorate these concerns. The fact that SSS can decrease current from both human and fly alpha containing complexes makes me suspicious that the SSS effect is indirect via some effect on trafficking/sorting in general or perhaps increased K channel expression. No data on RMP etc. are presented. Many more controls and experiments would be needed to make this cleanly interpretable (e.g., does SSS block current from other unrelated ionotropic receptors? Is there physical interaction of RYE and SSS? Co-expression of the point mutant RYE – is it really DN? Etc.).

It is also interesting that SSS seems to promote SHAKER expression yet suppress RYE. How do the authors fit this into a global model? They casually say its sleep promoting role is dominant but how would this really work? One thing we are never shown is where RYE is expressed in the brain. What is the pattern of *rye*-GAL4? Does this shed any light on the issue of SSS function? Since SHAKER is basically all over the brain perhaps RYE is more tightly localized and this might allow dissection of differential local effects of SSS on activity?

*Reviewer 2*
*comments:*

This paper presents studies that are truly a break-through for the sleep field – the identification of a mutant (*rye*) in which the affected protein is clearly part of the sleep homeostat (process S), which responds to increasing periods of wakefulness by producing a proportional increase in sleep drive. While the existence of such a homeostat is well supported by sleep behavior, the molecular basis has remained mysterious, and therefore this manuscript presents an important entry point for such a molecular analysis. The experiments are thorough, well controlled and convincing. The mutation affects a subunit for the nicotinic acetyl choline receptor, and the fact that a protein identified as mutated in a previously isolated short-sleeping mutant (*sss*) is shown to negatively regulate RYE is important because it suggests that the genetic analysis of sleep in *Drosophila* may be identifying some mutants in a single genetic pathway – something that has not been clear thus far. One perhaps could have hoped that this manuscript would also reveal the signals that lead to upregulation of RYE (i.e., more about the pathway for the sleep homeostat), but this will undoubtedly require more years of work by multiple labs.

The data supporting a role for RYE in the sleep homeostat are very compelling, but unless I'm missing something it seems to me that RYE expression is more likely to be triggered by an upstream S process than to be a direct part of the actual S process. For one thing, the phase for RYE expression is not right. A factor which directly responds to S should build up during wakefulness and dissipate during sleep, but RYE accumulates when the flies are asleep and only dissipates towards the end of sleep. While this feature of RYE expression alone might suggest that RYE accumulates during sleep to measure sleep recovery and promote waking, the short-sleep phenotype of the mutant and the elevated levels of RYE in various short-sleep mutants and under conditions of sleep deprivation are not consistent with this alternative (so it does seem that RYE is part of the process S). Another interesting feature is that RYE overexpression in a wild type genetic background does not alter sleep, which is inconsistent with a factor that directly drives the S process (this should elevate or phase advance sleep). The authors note this paradox and postulate that RYE overexpression may not be sufficient to induce sleep, or may not produce higher levels of RYE in the relevant cells (perhaps overexpressed proteins is not stable in these cells). Along these lines, perhaps RYE can only accumulate when the S process reaches a threshold that triggers sleep, and then RYE remains high until sleep debt is dissipated. Consideration of these points needs to be added to the Discussion; if nothing else, there needs to be more discussion of the phase of RYE expression and its relationship to its postulated role in the sleep homeostat.

*Reviewer 3*
*comments:*

This manuscript by Shi et al. describes sleep phenotypes of a very interesting mutation in *Drosophila* named redeye (*rye*). The authors do a very commendable job of isolating, identifying, characterizing and defining this mutation and in so doing further establish *Drosophila* as a powerful platform to study the biochemical and genetic basis of sleep. The authors convincingly show that loss of Rye leads to a short-sleeping phenotype that can be rescued partially by adding back Rye in a domain circumscribed by a newly generated Rye-GAL4 driver line and weakly phenocopied using RNAi. What is most interesting though is that the Rye protein (assayed using newly generated Rye antibodies) cycles during the day and is most prominently present just before sleep onset, is elevated by sleep deprivation and in several short-sleeping mutants such as sleepless, insomniac and fumin, but is not controlled by the clock machinery. These are hallmarks of a true “somnogen”; however, Rye fails in one test, i.e., artificially increased expression of Rye does not promote sleep. Even so, it seems clear that as the authors suggest, Rye is part of the sleep homeostat and is necessary but not sufficient for sleep drive. This is a very interesting study and is suitable for publication in *eLife*. One area that could have been investigated a bit more deeply is that of the tissue-specificity of Rye. The authors use the Rye-Gal4 and yet suggest that the expression pattern of this GAL4 line may not reflect that of endogenous Rye expression. Given the large number of extant GAL4 lines and some understanding of the sleep circuitry in *Drosophila*, it might have been instructive to test a few of these (e.g., pan neuronal, dopaminergic neurons, fan shaped body etc) to define the tissue-restricted requirement of Rye, if any. This could be most simply achieved using RNAi (though the phenotypes are arguably weaker than Rye mutants), selective rescue of Rye phenotypes using the UAS-Rye transgene, or perhaps even attempting to test if strong Rye over-expression in neuronal subsets might induce sleep (due to either stronger over-expression than Rye-GAL4 or a circuit-specific effect of Rye elevation). A minor point is the difference in the timing of the first peak of Rye between wild type and clock mutants (Figure 6). Do the authors think this is relevant or is the peak of Rye temporally more loosely defined than, say, clock proteins? Overall, this is an excellent manuscript and one of the few that describes a protein that is clearly correlated with sleep drive. Mechanistic associations with sleepless and channel activity further highlight the importance of this protein in sleep regulation.

---

## [Author Response]

*The consensus of the reviewers was that the identification of* rye *is an important and novel contribution. To our knowledge there are no other protein markers that have been reported that are independent of the clock and external conditions and respond only to sleep drive. RYE levels therefore potentially represent an important new biomarker for sleep deprivation. The reviewers also felt that the data on the interaction of* rye *and* sss *were the weakest part of the paper and that without deeper investigation the precise interpretation of those experiments was difficult. While it would be satisfying to have mechanism, this study is still quite important without a full understanding of the interaction of* rye *and* sss*. We would therefore recommend publication of a suitably revised paper. The reviews contain a number of minor things that will help the authors to make the paper stronger, but the main concerns/suggestions are enumerated below*.

We appreciate the reviewers’ remarks regarding the novelty and importance of the work. The comment pertaining to the interaction between *rye* and *sss* is addressed below, and in detail in the response to Reviewer 1.

*1) Eliminate*
Figure 5
*or cut back on the mechanistic claims and give a much fuller discussion of the caveats surrounding the data (see Reviewer 1 comments)*.

Because the interaction between *rye* and *sss* (Figure 5) represents a connection to a known sleep-regulating gene, and two very different assays (genetic and electrophysiology) are consistent in indicating evolutionarily conserved repression of nAchRs by SSS-like molecules, we chose to leave these data in the manuscript. We note too that Reviewer2 really liked this part of the manuscript. We did, however, conduct additional tests, and the new data (added to the manuscript) strengthen the case for genetic interaction. In addition, our colleague Bill Joiner recently obtained results indicating regulation of nAcHr by SSS (personal communication). Nevertheless, we acknowledge the reviewer’s concern, and have toned down the claims based upon these data (they are no longer mentioned in the Abstract) and we have also acknowledged the caveats in the Discussion.

*2) More balanced discussion of the issue of phase and whether* rye *is downstream of S or actually part of the homeostat (see Reviewer 2 comments). The overexpression data suggest the former*.

We agree that RYE is likely downstream of the homeostat. However, RYE is not a simple readout or output of the homeostat (S) as it is directly controlled by S, and functions to maintain the sleep process. We now discuss these issues in the text.

*3) Discussion of the issues surrounding the variability of the western blots (see Reviewers 1 and 3 comments)*.

We have added this (see also the Response to Reviewers 1 and 3). One possibility is that Process S cycles, but not with very precise timing because sleep and wake states can be influenced by many factors. The variable phase of RYE expression during an LD cycle may reflect the inaccurate timing of process S.

*4) More information on* rye *expression (or GAL4 pattern) and localization of requirement would strengthen the paper, especially if*
Figure 5
*is eliminated. If such data exist it would be good to include them, but we would not require new experiments to be done*.

We have included the ryeGAL4>GFP expression pattern in the brain. As mentioned in the initial submission, the expression pattern is broad, and does not implicate any specific structure in sleep regulation. We are mapping relevant cells, but thus far do not have any functional data.

Reviewer 1 comments:

*This is a very interesting study. The authors used the power of genetics to identify a novel sleep regulator: RYE. The fact that RYE levels are apparently controlled by sleep drive and not the clock make this a very novel and important discovery. I believe this may be the first molecule of this class and for that reason I am very enthusiastic about this paper*.

We thank the reviewer for these comments.

*Unfortunately, I think that the data that go beyond the characterization of* rye *are not very strong and are in many cases not interpretable in the simple way that the authors try to suggest. I almost wonder if the* sss *connection should not just be taken out of the paper until the authors have more data and can present a believable story. The isolation of the* rye *gene could stand alone as a brief communication. As it is the SSS stuff just muddies the waters. Perhaps if they added more detail about how the mutants are resistant to SD to the basic story*?

As noted above, we would prefer to leave in the *sss-rye* interaction data, which are now strengthened by additional genetic experiments, and are supported by data from Bill Joiner’s laboratory. We have also added a bit more discussion of the sleep deprivation experiments, although these are still not included as we did not feel we could draw any strong conclusions about them (we’re happy to include them if the reviewers deem otherwise).

*I am not convinced that there is a specific interaction between SSS and RYE that regulates sleep in WT animals. First, the authors claim that the* sssTg *lines have no effects on sleep, but they show data that the sleep levels in these lines are equivalent to* rye/+*, which has reduced sleep. The fact that now combining them results in even less sleep suggests a simple additive and independent effect of each manipulation on sleep. Second, there are no data on RNAi of RYE + OE of SSS. It is possible that the SSS effect is specific to the mutant allele because it has neomorphic function and that SSS may not regulate WT RYE*.

We believe that the apparent decrease in sleep in some *sssTg* lines (*sssTG2/+* and *sssTG3/+*) resulted from small sample sizes (n=8 and n=4, respectively). Please note that *sssTG1* did not reduce sleep in wild type, and yet this also reduced sleep in a *rye*/+ background. Nevertheless, we repeated the experiments with *sssTG2* and *sssTG3,* so as to increase the N*,* and pooled the old and new data. As shown in the revision, *sssTG2/+* (n=24) and *sssTG3/+* (n=12) have a sleep length comparable to *sssTG1/+* and to the *iso* stock. We conclude that the reduction of sleep in rye/ *sssTG* transhets is due to genetic interaction rather than additive effects.

It is true that the interaction could reflect a neomorphic function of the new mutant allele, and we have included this caveat in the discussion. As *rye* RNAi had a weak phenotype, we did not use it for additional experiments, but rather focused on the allele from the screen.

Finally, we note that interactions between SSS and nAchRs have been independently discovered by Bill Joiner (personal communication), so there is now increasing evidence to support such interactions.

*The oocyte data are not deep enough to ameliorate these concerns. The fact that SSS can decrease current from both human and fly alpha containing complexes makes me suspicious that the SSS effect is indirect via some effect on trafficking/sorting in general or perhaps increased K channel expression. No data on RMP etc. are presented. Many more controls and experiments would be needed to make this cleanly interpretable (e.g., does SSS block current from other unrelated ionotropic receptors? Is there physical interaction of RYE and SSS? Co-expression of the point mutant RYE – is it really DN? Etc.)*.

These are all valid points. We attempted to express RYE in S2 cells to test interactions with SSS, but, as noted by others previously for nAChR subunits, we could not get consistent expression. Thus, we could not perform co-immunoprecipitation assays. We have included this and other caveats in the Discussion. We have also deleted mention of the interaction data from the Abstract, and toned down our conclusions. However, we believe it is worth documenting the interaction in the manuscript, given that the genetic experiments are now stronger and the genetic and electrophysiology experiments are consistent in indicating repression of RYE by SSS. We believe the data will be of interest to many researchers, in that they indicate a conserved mechanism for regulation of nAchRs such as RYE by SSS-like molecules (given the known Lynx1-nAchR data in mammals). As noted above, Bill Joiner has also obtained evidence for regulation of nAchRs by SSS (personal communication).

*It is also interesting that SSS seems to promote SHAKER expression yet suppress RYE. How do the authors fit this into a global model? They casually say its sleep promoting role is dominant but how would this really work? One thing we are never shown is where RYE is expressed in the brain. What is the pattern of* rye*-GAL4? Does this shed any light on the issue of SSS function? Since SHAKER is basically all over the brain perhaps RYE is more tightly localized and this might allow dissection of differential local effects of SSS on activity*?

We have added more discussion of these issues. As *sss* mutants have a stronger sleep phenotype than *Shaker,* it is not surprising that *sss* also has other functions. For instance, SSS may normally promote sleep, not only by promoting Shaker activity, but also by inhibiting activity of various nAchRs, most of which are likely wake-promoting (RYE is clearly an exception). We have included the expression pattern of rye-GAL4 driven GFP. It is quite broad, so it does not reveal why the effect of SSS on RYE is masked in *sss* mutants. Possibly, co-expression of RYE and SSS occurs in a limited number of sleep-regulating cells.

Reviewer 2 comments:

*This paper presents studies that are truly a break-through for the sleep field - the identification of a mutant (*rye*) in which the affected protein is clearly part of the sleep homeostat (process S), which responds to increasing periods of wakefulness by producing a proportional increase in sleep drive. While the existence of such a homeostat is well supported by sleep behavior, the molecular basis has remained mysterious, and therefore this manuscript presents an important entry point for such a molecular analysis. The experiments are thorough, well controlled and convincing. The mutation affects a subunit for the nicotinic acetyl choline receptor, and the fact that a protein identified as mutated in a previously isolated short-sleeping mutant (*sss*) is shown to negatively regulate RYE is important because it suggests that the genetic analysis of sleep in Drosophila may be identifying some mutants in a single genetic pathway - something that has not been clear thus far. One perhaps could have hoped that this manuscript would also reveal the signals that lead to upregulation of RYE (i.e., more about the pathway for the sleep homeostat), but this will undoubtedly require more years of work by multiple labs*.

We thank the reviewer for these very complimentary remarks.

*The data supporting a role for RYE in the sleep homeostat are very compelling, but unless I'm missing something it seems to me that RYE expression is more likely to be triggered by an upstream S process than to be a direct part of the actual S process. For one thing, the phase for RYE expression is not right. A factor which directly responds to S should build up during wakefulness and dissipate during sleep, but RYE accumulates when the flies are asleep and only dissipates towards the end of sleep. While this feature of RYE expression alone might suggest that RYE accumulates during sleep to measure sleep recovery and promote waking, the short-sleep phenotype of the mutant and the elevated levels of RYE in various short-sleep mutants and under conditions of sleep deprivation are not consistent with this alternative (so it does seem that RYE is part of the process S). Another interesting feature is that RYE overexpression in a wild type genetic background does not alter sleep, which is inconsistent with a factor that directly drives the S process (this should elevate or phase advance sleep). The authors note this paradox and postulate that RYE overexpression may not be sufficient to induce sleep, or may not produce higher levels of RYE in the relevant cells (perhaps overexpressed proteins is not stable in these cells). Along these lines, perhaps RYE can only accumulate when the S process reaches a threshold that triggers sleep, and then RYE remains high until sleep debt is dissipated. Consideration of these points needs to be added to the Discussion; if nothing else, there needs to be more discussion of the phase of RYE expression and its relationship to its postulated role in the sleep homeostat*.

These are all great points. We agree that RYE could be immediately downstream of process S. RYE levels may be higher in short sleepers because those mutants have defects in further downstream regulators (e.g., dopamine pathways). We have added discussion of these possibilities to the manuscript.

Reviewer 3 comments:

*This manuscript by Shi et al. describes sleep phenotypes of a very interesting mutation in* Drosophila *named redeye (*rye*). The authors do a very commendable job of isolating, identifying, characterizing and defining this mutation and in so doing further establish* Drosophila *as a powerful platform to study the biochemical and genetic basis of sleep. The authors convincingly show that loss of Rye leads to a short-sleeping phenotype that can be rescued partially by adding back Rye in a domain circumscribed by a newly generated Rye-GAL4 driver line and weakly phenocopied using RNAi. What is most interesting though is that the Rye protein (assayed using newly generated Rye antibodies) cycles during the day and is most prominently present just before sleep onset, is elevated by sleep deprivation and in several short-sleeping mutants such as sleepless, insomniac and fumin, but is not controlled by the clock machinery. These are hallmarks of a true “somnogen”; however, Rye fails in one test, i.e., artificially increased expression of Rye does not promote sleep. Even so, it seems clear that as the authors suggest, Rye is part of the sleep homeostat and is necessary but not sufficient for sleep drive. This is a very interesting study and is suitable for publication in eLife. One area that could have been investigated a bit more deeply is that of the tissue-specificity of Rye. The authors use the Rye-Gal4 and yet suggest that the expression pattern of this GAL4 line may not reflect that of endogenous Rye expression. Given the large number of extant GAL4 lines and some understanding of the sleep circuitry in* Drosophila*, it might have been instructive to test a few of these (e.g., pan neuronal, dopaminergic neurons, fan shaped body etc) to define the tissue-restricted requirement of Rye, if any. This could be most simply achieved using RNAi (though the phenotypes are arguably weaker than Rye mutants), selective rescue of Rye phenotypes using the UAS-Rye transgene, or perhaps even attempting to test if strong Rye over-expression in neuronal subsets might induce sleep (due to either stronger over-expression than Rye-GAL4 or a circuit-specific effect of Rye elevation)*.

As the reviewer notes, the RNAi phenotype is weak, so we did not use this as an assay to identify the relevant circuit. We are attempting to map the circuit through other means, but do not have any conclusions thus far.

*A minor point is the difference in the timing of the first peak of Rye between wild type and clock mutants (*Figure 6*). Do the authors think this is relevant or is the peak of Rye temporally more loosely defined than, say, clock proteins? Overall, this is an excellent manuscript and one of the few that describes a protein that is clearly correlated with sleep drive. Mechanistic associations with sleepless and channel activity further highlight the importance of this protein in sleep regulation*.

The peak of RYE is indeed more loosely defined than that of clock proteins. The phase is even looser in short-sleeping mutants and in *Clk*^*jrk*^*.* It is possible that these data reflect the nature of Process S, which is perhaps more flexible/sloppy than Process C (circadian control).